# ICESat-2 for Coastal MSS Determination—Evaluation in the Norwegian Coastal Zone

**Matea Tomić** [1,*] and **Ole Baltazar Andersen** [2]

1   Department of Geomatics, Faculty of Science and Technology (RealTek),
    Norwegian University of Life Sciences (NMBU), N-1432 Ås, Norway
2   DTU Space, Technical University of Denmark, Elektrovej 328, 2800 Kongens Lyngby, Denmark;
    oa@space.dtu.dk
*   Correspondence: matea.tomic@nmbu.no

**Abstract:** Radar satellite altimeters enable the determination of the mean sea surface to centimeter accuracy, which can be degraded in coastal areas because of the lack of valid altimetry observations due to land contamination and the altimeter footprint size. In 2018, the National Aeronautics and Space Administration launched ICESat-2, a laser altimetry mission equipped with the Advanced Topographic Laser Altimeter System, providing measurements every 0.7 m in the along-track direction. Taking into account the complexity of the Norwegian coastline, this study aims to evaluate coastal observations from ICESat-2 in order to use it to update the existing mean sea surface for Norway, NMBU18. We, therefore, determined the mean sea surface using only ICESat-2 observations and compared it with mean sea level observations from 23 permanent tide gauges along the entire coast and 21 temporary tide gauges in Norway's largest and deepest fjord, Sognefjorden. We also included two global mean sea surface models and NMBU18 for comparison. The results have shown that ICESat-2 is indeed able to provide more valid observations in the coastal zone, which can be used to improve the mean sea surface model, especially along the coast.

**Keywords:** coastal sea level; ICESat-2; laser altimetry; mean sea surface; satellite altimetry; tide gauges

## 1. Introduction

The mean sea surface (MSS) represents an important parameter in geodesy and physical oceanography. It is created by the time-averaging of the sea surface heights (SSHs) observed by radar satellite altimetry over a long time period [1,2]. Along with the increased number of radar altimetry missions, improved onboard instrumentation, as well as signal processing, the accuracy of altimetry-based MSS models has also improved. Nevertheless, the main drawback of radar altimetry is the degraded performance in the coastal zone due to uncertainties in range and geophysical corrections, and the size of the altimeter footprint [3–5].

In September 2018, the National Aeronautics and Space Administration (NASA) launched the Ice, Cloud, and Land Elevation Satellite-2 (ICESat-2), a laser altimetry mission equipped with the sole instrument, the novel photon-counting Advanced Topographic Laser Altimeter (ATLAS) [6,7]. Together with ATLAS, ICESat-2 is equipped with ancillary systems (GPS and star cameras) to measure the time a photon takes to make a round-trip from the satellite to Earth and to determine the photon's longitude, latitude, and height relative to the WGS84 ellipsoid [8]. ATLAS uses a low-pulse energy green (532 nm) laser with a footprint size of around 17 m which emits 10.000 laser pulses per second, with about 20 trillion photons in each pulse. This fast pulse rate enables ATLAS to take measurements every 0.7 m in the along-track direction. The single output laser pulse is split into three pairs of beams, with a distance of 3.3 km between each pair. Within each pair, a strong and

weak beam is located at the two sides of the reference ground track (RGT) at a distance of 90 m (see Figure 1) [8].

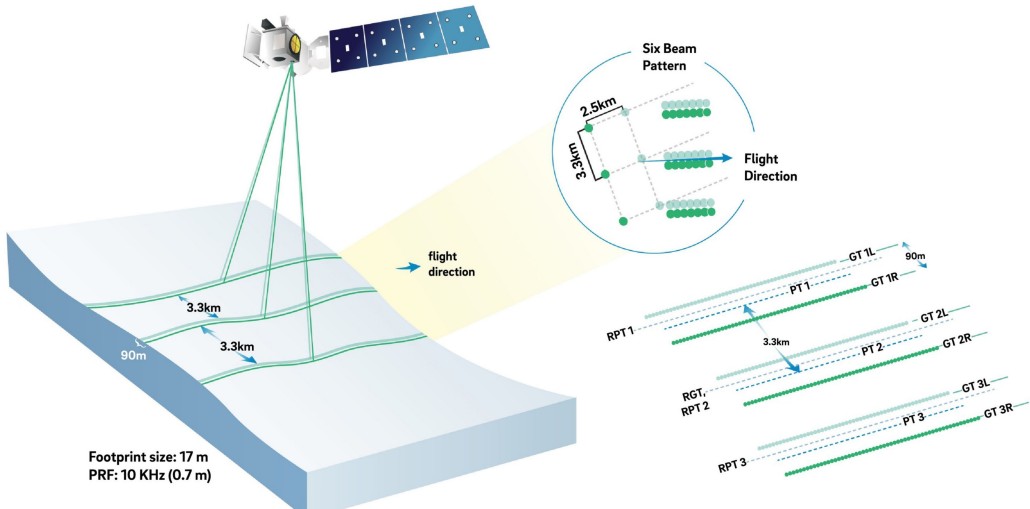

**Figure 1.** The graphic representation of the six-beam pattern from the ATLAS instrument and its measurement technique. Figure reprinted from [9]. Copyright 2023, with permission from Elsevier.

Geolocated photons are provided within ATLAS/ICESat-2 L2A Global Geolocated Photon Data (ATL03) product, which is designed as a single source for all photon data and ancillary information needed for higher-level products [10,11].

The primary objectives of ICESat-2 are related to monitoring land ice elevation and sea ice changes. However, as the satellite collects measurements over all types of surfaces, particularly the ocean, there is potential for ICESat-2 to provide new insights into understanding ocean dynamics [12,13]. Thus, the ICESat-2 ocean data product, ATL12, has been developed to provide sea surface heights along the track at variable length scales over cloud-free regions. The ATL12 product is derived from the ATL03 product [14].

ICESat-2 has a nominal footprint of 17 m and an along-track resolution of 0.7 m, compared to radar altimeter footprints of several kilometers [15]. Compared with new-generation radar altimetry missions equipped with Synthetic Aperture Radar (SAR) with along-track resolution of 300 m (Sentinel-3 and Cryosat-2), ICESat-2 is able to retrieve more observations closer to the coast and even inside fjords without compromising its accuracy [13] (Figure 2).

The disadvantage of ICESat-2 is its repeat cycle of 91 days, compared to radar altimetry missions (with the exception of CryoSat-2), which have repeat cycles of 10 to 35 days [16]. Another disadvantage is the sensitivity of the laser to weather conditions, particularly the presence of clouds, which can reduce the reflectivity and therefore the number of laser observations.

In this study, we want to test the potential of the ICESat-2 ATL12 product to improve the existing coastal mean sea surface for Norway, NMBU18 [17]. The coastal mean sea surface plays a crucial role in connecting the open sea mean sea surface (MSS) with the tide gauges, serving as the essential element for unifying the vertical reference frames [17]. The Norwegian coastline is well known for its complexity, and retrieving valid radar altimetry observations close to the coast, especially inside fjords, can be very challenging as shown in previous studies by [18–20]. Furthermore, the lack of altimetry observations is reflected in the error field of NMBU18, which shows increased errors near the coast and in fjords [17]. This study, therefore, represents the first evaluation of ICESat-2 along the Norwegian coast. Due to its small footprint, ICESat-2 has the potential to fill the gap between coastal tide gauges and the open sea, to fill the fjords with sea surface height measurements and to complement tide gauge measurements. To test the possibility of incorporating ICESat-2 observations into a new mean sea surface model, we constructed a mean sea surface model

using only ICESat-2 ATL12 data, and we validated it with the NMBU18 and state-of-the-art global MSS models. To assess whether ICESat-2 can provide more valid observations inside fjords and contribute to a more detailed MSS model for Norway, we also compared the ICESat-2 ATL12 product with observations from the Norwegian network of permanent tide gauges along the entire coast and temporary tide gauges in Sognefjorden, the largest and deepest fjord in Norway.

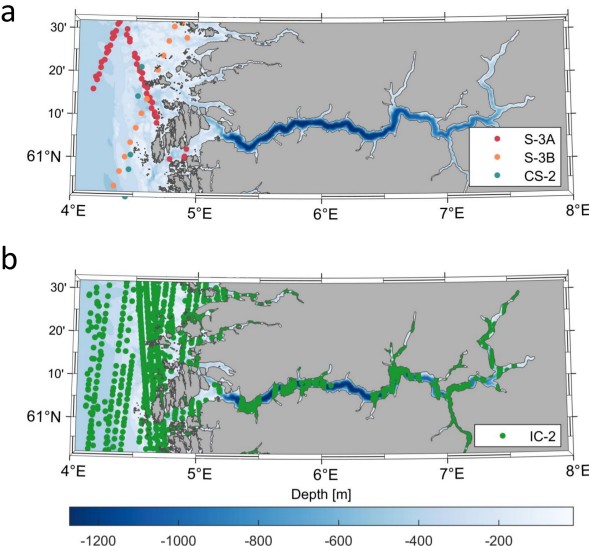

**Figure 2.** Locations of Sentinel-3/CryoSat-2 (**a**) and ICESat-2 (**b**) observations inside Sognefjorden during 2020. Bathymetry is taken from the 2021 General Bathymetric Charts of the Oceans (GEBCO) [21].

## 2. Metocean Conditions of the Study Area

Norway is located in the western part of the Scandinavian peninsula with mountainous terrain, thousands of islands, and narrow, deep fjords spread all along the coastline. The Norwegian coast is formed along the Skagerrak Strait, North Sea, Norwegian Sea, and Barents Sea, and their hydrography and bathymetry have an impact on the overall coastal current pattern [22].

In the Skagerrak, the northbound Baltic current follows the Norwegian coast westwards and continues as the Norwegian Coastal Current (NCC), which dominates sea level variability along most of the Norwegian coast [23]. NCC flows north along the coast of Norway and it consists of Norwegian coastal water (NCW) and Atlantic water (AW). NCW appears close to the coast and it is characterized by low salinity and cold temperature, as a result of large amounts of freshwater run-off from rivers and fjords. AW is transported into the Norwegian Sea by the North Atlantic Current (NAC), which is the reason why the coastal areas, even at 68°N, are ice-free during winter [22].

Besides currents, winds can significantly change the water level. By pushing water towards the shore, they can create an excess of water concentrated along the coast. Along the Norwegian coast, this is mainly caused by winds from the south and west. Besides that, strong winds in the North Sea can create long waves which cause dramatically high sea levels, even though the wind is calm locally. Strong winds, together with low atmospheric pressure, contribute to creating storm surges. When a storm surge coincides with so-called high astronomical tides, extremely high water levels occur [24].

Tides also are a dominant contributor to sea level variations. In Nordic seas, the tidal wave is dominated by the semi-diurnal constituents [22,25]. The tidal amplitude is only a few decimeters along the south coast of Norway. With increasing latitude, the tidal range also increases by more than one meter in northern Norway. Strong tidal currents are common in the northern parts of Norway, e.g., the world-famous Moskenstraumen tidal current near the Lofoten archipelago [22]. Due to the smaller tides in the south,

the weather-related sea level variations dominate in the south, whereas tidal variations dominate in the north.

The general representation of the total observed water level $\mathbf{X}(t)$ which varies with time can be expressed as:

$$\mathbf{X}(t) = Z_0(t) + \mathbf{T}(t) + \mathbf{S}(t), \tag{1}$$

where $Z_0(t)$ is the mean sea level which slowly changes with time, $\mathbf{T}(t)$ is the tidal part of the variation and $\mathbf{S}(t)$ is the meteorological part of the variation. In addition, the tidal and meteorological (surge) components of the series $\mathbf{X}(t)$ have a condition that they are statistically independent. Thus, the sum of their individual variances gives us the variance in the total observed series [26]:

$$\sum_{k=1}^{M}(\mathbf{X}(k\Delta t) - Z_0)^2 = \sum_{k=1}^{M}\mathbf{T}^2(k\Delta t) + \sum_{k=1}^{M}\mathbf{S}^2(k\Delta t) \tag{2}$$

In Section 3.2, we provide more information about tide gauges and the various contributions to sea level variance for each tide gauge.

## 3. Data and Methods

### 3.1. ICESat-2 ATL12 Data

We used the ICESat-2 ATL12 Ocean Surface Height Version 5 product provided by the National Snow and Ice Data Center, which is updated quarterly in accordance with the 91-day repeat time of the ICESat-2 satellite [14]. The ATL12 product contains sea surface heights (SSH) relative to the WGS84 reference ellipsoid at various length scales derived from geolocated, time-tagged photon heights from the ATLAS/ICESat-2 L2A Global Geolocated Photon Data, along with other parameters. The ATL12 algorithm calculates the SSH from accumulated photon returns over 7 km of orbit, or from accumulated 8000 photons. More details about the ATL12 algorithm can be found in [8,14].

As an output, the ATL12 adaptive algorithm provides SSHs on length scales between 0.7 m and 7 km, depending on conditions and cloud cover. Before further analysis, the geoid model (EGM2008 in the mean tide system [27]) is subtracted from the SSH to reduce the signal amplitude, i.e., the analysis is performed in terms of Dynamic Ocean Topography (DOT) [8,14,28]. EGM2008 is preferred due to its true global coverage, whereas global MSS models, which are based on radar altimetry observations, suffer from degradation in complicated coastal zones, as seen in Figure 2.

The SSH variable in the ATL12 product is corrected for tidal and atmospheric effects. However, in coastal areas not covered by the GOT4.8 ocean tide model (<25 km off the coast), we have used regional ocean tide predictions provided by the Norwegian Mapping Authority (NMA).

### 3.2. Tide Gauge Observations

Observations from 23 permanent tide gauges from the Norwegian coast and from 21 temporary tide gauges around Sognefjorden were provided by NMA (Figure 3).

Their locations, names, identification codes as well as observation periods, are provided in Tables 1 and 2.

Tables also show the values of variances of observed sea level before (Var_obs_SL) and after (Var_SL) applying tidal correction, as well as variance of sea level observations from ICESat-2, computed for detided observations within 10 km (15 km inside Sognefjorden) around each tide gauge.

From Table 1, it can be seen that the observed sea level variance varies from roughly 3 cm$^2$ in the south to around 65 cm$^2$ in the north. This corresponds to sea level variations ranging from 15 cm to 80 cm in terms of standard deviations. In all cases, the tides contribute at least 80% of the variance in sea level, especially in northern Norway, compared to southern Norway, where tidal contribution to the variance of the total observed sea level is significantly smaller. The standard deviation of the residual sea level variance is

generally modest with values smaller than 25 cm almost everywhere. Hence, it is important to remove the effect of tidal variability before computing the mean sea surface.

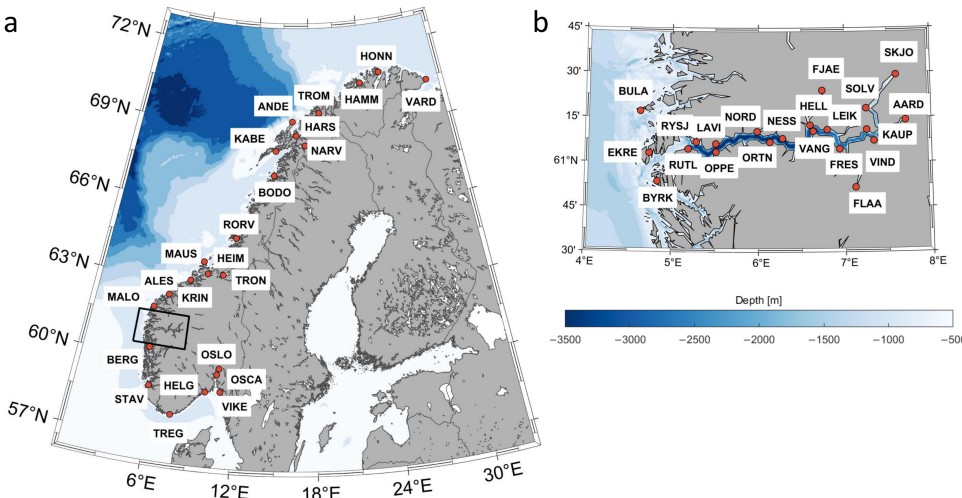

**Figure 3.** Locations of permanent tide gauges along the Norwegian coast (**a**) and temporary tide gauges inside Sognefjorden (**b**) used in this study. Bathymetry is taken from the 2021 General Bathymetric Charts of the Oceans (GEBCO) [21].

**Table 1.** Locations of permanent tide gauges for the period 2018–2022, and the values of variances for different contributors to sea level observations expressed in cm$^2$. Letters OS and F denote whether the tide gauge is located towards the open sea (OS) or inside fjords (F).

| Tide Gauge | ID | $\lambda$ [°] | $\phi$ [°] | Var_obs_SL | Var_T | Var_SL | Var_IC-2 |
|---|---|---|---|---|---|---|---|
| Vardø | VARD (OS) | 31.104 | 70.375 | 63.4 | 60.7 | 2.7 | 2.3 |
| Honningsvåg | HONN (OS) | 25.973 | 70.980 | 46.7 | 44.1 | 2.7 | 5.0 |
| Hammerfest | HAMM (OS) | 23.683 | 70.665 | 48.9 | 46.1 | 2.8 | 3.4 |
| Tromsø | TROM (F) | 18.961 | 69.647 | 44.8 | 41.9 | 2.9 | 2.8 |
| Andenes | ANDE (OS) | 16.135 | 69.326 | 29.5 | 26.4 | 3.2 | 2.8 |
| Harstad | HARS (F) | 16.548 | 68.801 | 32.3 | 29.4 | 2.9 | 4.2 |
| Narvik | NARV (F) | 17.426 | 68.428 | 64.4 | 60.5 | 3.8 | 2.0 |
| Kabelvåg | KABE (OS) | 14.482 | 68.213 | 56.0 | 52.2 | 3.8 | 2.7 |
| Bodø | BODO (OS) | 14.391 | 67.288 | 49.5 | 45.8 | 3.7 | 2.5 |
| Rørvik | RORV (OS) | 11.230 | 64.859 | 40.8 | 37.3 | 3.5 | 2.2 |
| Mausund | MAUS (OS) | 8.665 | 63.869 | 35.4 | 32.4 | 3.0 | 3.4 |
| Heimsjø | HEIM (F) | 9.102 | 63.425 | 39.7 | 36.3 | 3.4 | 2.3 |
| Trondheim | TRON (F) | 10.392 | 63.436 | 54.3 | 51.3 | 3.0 | 3.8 |
| Kristiansund | KRIN (OS) | 7.734 | 63.114 | 31.2 | 27.7 | 3.6 | 2.5 |
| Ålesund | ALES (OS) | 6.152 | 62.469 | 26.7 | 23.1 | 3.6 | 1.5 |
| Måløy | MALO (OS) | 5.113 | 61.934 | 23.6 | 20.3 | 3.3 | 0.6 |
| Bergen | BERG (F) | 5.321 | 60.398 | 14.8 | 12.3 | 2.5 | 0.6 |
| Stavanger | STAV (OS) | 5.730 | 58.974 | 4.3 | 1.7 | 2.6 | 1.3 |
| Tregde | TREG (OS) | 7.555 | 58.006 | 2.7 | 0.6 | 2.1 | 1.6 |
| Helgeroa | HELG (OS) | 9.856 | 58.995 | 4.1 | 0.9 | 3.2 | 1.6 |
| Oscarsborg | OSCA (F) | 10.605 | 59.678 | 6.1 | 1.4 | 4.7 | 2.9 |
| Oslo | OSLO (F) | 10.735 | 59.909 | 6.6 | 1.5 | 0.5 | 1.1 |
| Viker | VIKE (O) | 10.950 | 59.036 | 5.1 | 1.0 | 4.1 | 2.1 |

**Table 2.** Location and observation periods of temporary tide gauges inside Sognefjorden (S)., and the values of variances for different contributors to sea level observations expressed in cm$^2$.

| Tide Gauge | ID | $\lambda$ [°] | $\phi$ [°] | Obs. Period | Var_obs_SL | Var_SL | Var_T | Var_IC-2 |
|---|---|---|---|---|---|---|---|---|
| Bulandet | BULA | 4.633 | 61.286 | Apr 2018–Jun 2019 | 17.6 | 15.2 | 2.4 | 2.9 |
| Ekrevika | EKRE | 4.738 | 61.052 | Apr 2018–Jun 2019 | 16.7 | 14.4 | 2.3 | 0.3 |
| Byrknes | BYRK | 4.840 | 60.894 | Apr 2018–Jun 2019 | 15.3 | 13.0 | 2.2 | 1.5 |
| Rutledal | RUTL | 5.190 | 61.075 | May 2018–Jun 2019 | 18.3 | 15.8 | 2.5 | 0 |
| Rysjedalsvika | RYSJ | 5.283 | 61.114 | May 2018–Jun 2019 | 18.1 | 17.7 | 2.3 | 1.0 |
| Oppedal | OPPE | 5.510 | 61.057 | May 2018–Jun 2019 | 18.2 | 15.8 | 2.4 | 1.1 |
| Lavik | LAVI | 5.508 | 61.104 | May 2018–Jun 2019 | 18.9 | 16.0 | 2.9 | 2.0 |
| Nordeide | NORD | 5.9859 | 61.172 | May 2018–Oct 2019 | 18.9 | 16.1 | 2.8 | 1.9 |
| Ortnevik | ORTN | 6.133 | 61.113 | May 2018–Jan 2019 | 18.8 | 16.4 | 2.4 | 1.5 |
| Nessane | NESS | 6.279 | 61.134 | May 2018–Jun 20199 | 18.8 | 16.5 | 2.3 | 1.8 |
| Fjærland | FJAE | 6.739 | 61.402 | Jun 2018–Jun 2019 | 19.5 | 17.0 | 2.4 | 2.1 |
| Hella | HELL | 6.597 | 61.208 | May 2018–May 2020 | 19.4 | 16.6 | 2.8 | 1.6 |
| Vangsnes | VANG | 6.633 | 61.174 | Jun 2018–Jun 2019 | 19.2 | 16.9 | 2.3 | 0.8 |
| Leikanger | LEIK | 6.800 | 61.183 | Jun 2018–Oct 2020 | 19.3 | 16.7 | 2.6 | 1.6 |
| Fresvik | FRES | 6.939 | 61.074 | May 2018–Jun 2019 | 18.6 | 17.1 | 1.5 | 1.1 |
| Flåm | FLAA | 7.120 | 60.861 | Jun 2018–Jun 2019 | 19.4 | 17.3 | 2.1 | 1.0 |
| Kaupanger | KAUP | 7.252 | 61.182 | Jun 2018–Oct 2020 | 19.5 | 16.9 | 2.6 | 1.1 |
| Vindedalen | VIND | 7.336 | 61.121 | Jun 2018–Jun 2019 | 19.6 | 17.2 | 2.4 | 2.0 |
| Årdalstangen | AARD | 7.703 | 61.235 | Jun 2018–May 2020 | 20.0 | 2.9 | 17.1 | 2.2 |
| Solvorn | SOLV | 7.249 | 61.301 | Jun 2018–Jun 2019 | 19.6 | 17.2 | 2.1 | 4.0 |
| Skjolden | SKJO | 7.600 | 61.488 | Jun 2018–May 2020 | 20.3 | 17.2 | 3.1 | 4.5 |

Furthermore, it can be seen that the variance of ICESat-2 observations is slightly different in most places than the expected sea level variance observed by tide gauges. This is partly caused by the wave setup close to the coast. However, since ATL12 averages ATL03 observations over several kilometers this product is much less prone to the wave setup in the coastal zone [14,29].

In addition, values of variances of the observed sea level computed from tide gauges inside Sognefjorden show lower variations, varying between 15.3 cm$^2$ (Byrknes) and 20.3 cm$^2$ (Skjolden), as well as residual sea level variance (Table 2). However, variances of detided sea level observations from ICESat-2 show greater variability compared to variances computed from tide gauge observations.

ICESat-2 provides ellipsoidal heights in the international terrestrial reference frame (ITRF2014), and observations are given in the mean tide system. On the other hand, sea level measurements are related to the zero point of each tide gauge, as tide gauges provide observations relative to the land on which they are located. In order to make them consistent with altimetry observations, it is, therefore, necessary to calculate ellipsoidal sea level heights of tide gauge zero in ITRF2014 and to convert them from a tide-free to a mean tide system. This was conducted using the approach described in [30].

To maintain consistency with the altimetry data, ocean tide correction (OT) and Dynamic Atmospheric Correction (DAC) were applied to the tide gauge observations. Ocean tide predictions were downloaded using NMA's dedicated API (http://api.sehavniva.no, accessed on 19 January 2023), and DAC corrections were downloaded from the Archiving, Validation and Interpretation of Satellite Oceanographic data (AVISO) (https://www.aviso.altimetry.fr, accessed on 25 June 2023). In addition, as temporary tide gauges cover different time periods, reference corrections were applied for the 2018–2022 reference period, as described in [20].

### 3.3. Mean Sea Surface Models

We compared our MSS model derived from ICESat-2 observations (hereafter referred to as IC-2 MSS) with the following global multimission MSS models: DTU21 from the Technical University of Denmark (DTU) [31] and CNES_CLS15 from the Centre national d'études spatiales (CNES) [32], as well as the MSS model for the Norwegian coast, NMBU18 [17].

NMBU18MSS is based on 7 years (2010–2017) of SAR(In) data from Sentinel-3A, CryoSat-2 and Ka-band data from SARAL/AltiKa, covering the area from 0°E to 34°E and from 55.83°N to 73°N.

The mean sea surface MSS_CNES_CLS_15 is based on a 20-year period (1993–2012) of altimetry data from the following missions: Topex/Poseidon, Jason-1, Jason-2, Topex/Poseidon Tandem, ERS-2, Envisat, GFO, ERS-1 and CryoSat-2. It is based on the same remove/restore approach as NMBU18 [32] in terms of the Mean Dynamic Topography (MDT), where the geoid model was removed a priori and subsequently added after the MDT was determined.

DTU21MSS is the latest release of the global MSS models produced at DTU Space, calculated over the same 20-year averaging period from 1 January 1993 to 31 December 2012. It is implemented with an improved 10-year Cryosat-2 LRM+SAR+SARin record including retracked altimetry in the polar regions using the SAMOSA+ physical retracker via the ESA GPOD facility and incorporates altimetry data until the end of 2020 [31]. DTU21 is based on the same computational procedure as previous DTU MSS models, where the long-wavelength part of the MSS is computed from the time-averaged ERM (Exact Repeat Mission) data, and the GM (Geodetic Mission) data are subsequently added to compute the small-scale structures of the MSS [1,31].

All models are given as regular grids with a spatial resolution of 1/60° (1 min) (i.e., ~1.8 km). Since DTU21 and CNES_CLS15 are referenced to the Topex/Poseidon ellipsoid, datum transformation to the WGS84 ellipsoid was required to maintain consistency with the ICESat-2 observations. The datum transformation was performed using expressions adapted from [33]:

$$\Delta h = \frac{a'\left(1 - e'^2\right)}{\left(1 - e'^2 \sin^2 \varphi\right)^{3/2}} - \frac{a\left(1 - e^2\right)}{\left(1 - e^2 \sin^2 \varphi\right)^{3/2}} \tag{3}$$

where $\Delta h$ is the change in height at latitude $\varphi$ due to the change of the ellipsoid to the WGS84 datum. $a$ = 6,378,137 m and $e$ = 0.081819190842621 are the semi-major axis and the eccentricity of the WGS84 datum, while $a'$ = 6,378,136.3 m and $e'$ = 0.081819221456 are the semi-major axis and the eccentricity of the T/P ellipsoid, respectively.

As two of the models have a center period of 2003 and NMBU18 of 2014.5, we applied the correction for ongoing sea level rise by adding the corresponding values of sea level rise to each grid cell determined by [34].

### 3.4. MSS Determination from ICESat-2

Prior to any further analysis, it was essential to identify and remove outliers from the dataset used. The ICESat-2 ATL12 data were pre-processed in terms of sea level anomalies (SLAs), using the DTU21 mean sea surface model to reference SLAs from ICESat-2 ATL12 sea surface heights. In a first step, all SLAs deviating more than ±2 m from DTU21MSS were removed. In the next step, we performed the gross error search using a multiple *t*-test [18,35,36] within each ICESat-2 track. A detailed description can be found in [18]. The observations were then divided into 5 km × 5 km grid cells. Within each grid cell, the annual signal was estimated by least squares adjustment and removed [37]. The mean and standard deviation of each grid cell were then determined, and all values outside the 1.7 standard deviations of the mean were defined as outliers and removed. The outlier detection process resulted in a ~6% data rejection.

Afterwards, the filtered and processed observations were interpolated on the IC-2 mean sea surface model on a regular grid extending from 0°E to 34°E, and from 55.83°N to 73°N, with a 1 km resolution, thus having the same resolution and spatial coverage as NMBU18. For the interpolation method, we have used a simple natural-neighbor interpolation method [38].

## 4. Results and Discussion

The resulting MSS model, made only from ICESat-2 observations, is shown in Figure 4.

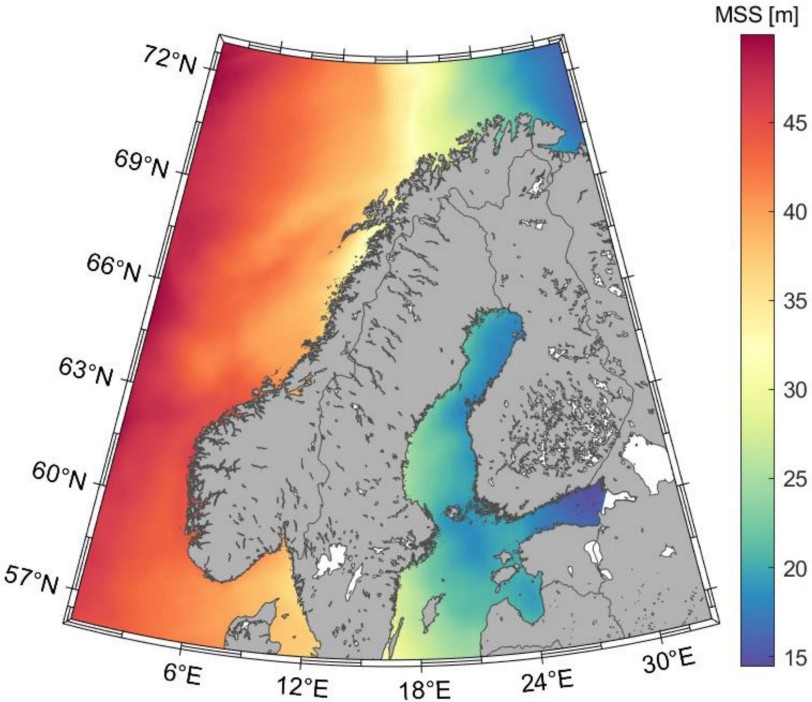

**Figure 4.** The mean sea surface for Norway for the period 2018–2022 from ICESat-2 observations.

As the main focus of this study is to investigate whether ICESat-2 data can be used to improve the mean sea surface in the Norwegian coastal zone, we compared the differences in mean sea level (MSL) from each tide gauge over the period 2018 to 2022 with the MSS value from the nearest grid cell of each model (Figures 5 and 6).

If we take a look at the performance of the MSS models along the entire coast (Figure 5), it can be seen that CNES_CLS15 performs the worst in comparison with the tide gauges, with standard deviations of 21.6 cm (Table 3). It shows large differences (>20 cm) with tide gauges located in Nordland and Trøndelag (NARV-TRON) and in Oslofjord (OSCA and OSLO). The highest differences (>30 cm) are visible at NARV, TRON, OSLO and BODO tide gauges. NARV, TRON, OSLO are located deep inside fjords, where NMBU18's error field shows large uncertainties, due to the lack of altimetry observations [17]. The new release of the DTU21 MSS model outperforms the existing MSS model for Norway, NMBU18, especially at the above-mentioned tide gauges: NARV, TRON, OSLO. However, DTU21 shows an extremely large difference in height with the TROM tide gauge (>30 cm), as well NMBU18 with the KABE tide gauge (>50 cm), where IC-2 MSS shows a big improvement and reduces the difference in height to less than 20 cm.

Compared to the other models, IC-2 MSS shows improvement in terms of the mean differences (−2.1 cm) and standard deviations (11.1 cm) compared to DTU21 (14.6 cm) and NMBU18 (16.0 cm). However, an extreme value of −20.4 cm was observed at the RORV tide gauge.

Looking back at Table 1, considering the maximum value of sea level variance (5.0 cm at Honningsvåg) and dividing that number by the average number of observations from ascending and descending tracks within the 10 km grid cell, the result contributes to an observation error of 3.6 cm ($5.0^2/\sqrt{36}$), assuming that the noise is white. Looking at the standard deviation of the mean sea surface within the closest grid cell (Figure 5), it can be seen that the error of the computed mean sea surface model is higher at some stations than expected since the total noise of the MSS will also contain other contributions. The grid cell closest to the RORV tide gauge has the highest value of the signal standard deviation of

almost 40 cm. Furthermore, the standard deviations of the ICESat-2 observations in the vicinity of land-confined tide gauges (TROM, NARV, TRON, OSCA, OSLO) tend to have similar values to observations close to open sea tide gauges.

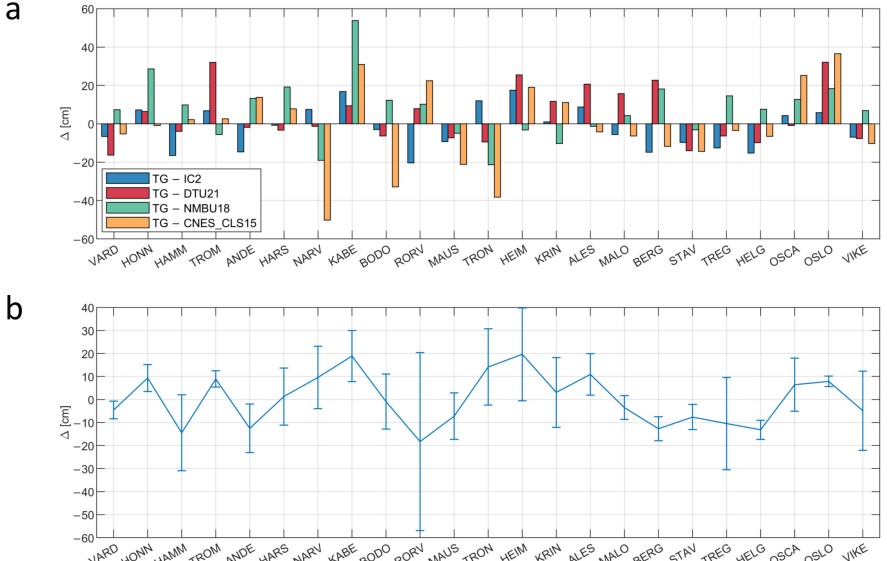

**Figure 5.** The height difference between permanent tide gauges and the IC-2 model (**a**). Signal standard deviations of the closest grid cell with available ICESat-2 observations (**b**).

**Table 3.** Statistics of differences between MSL observed by permanent tide gauges and MSS from altimetry.

| Δ | MIN [cm] | MAX [cm] | MEAN [cm] | STD [cm] |
|---|---|---|---|---|
| TG-IC2 | −20.4 | 17.5 | −2.1 | 11.1 |
| TG-DTU21 | −16.4 | 32.0 | 4.1 | 14.6 |
| TG-NMBU18 | −21.4 | 53.9 | 7.3 | 16.0 |
| TG-CNES_CLS15 | −50.3 | 36.6 | −1.5 | 21.6 |

Inside Sognefjorden, CNES_CLS15 shows better agreement with tide gauges in terms of the standard deviation (27.4 cm) than both NMBU18 (45.2 cm) and DTU21 (42.9 cm), see Table 4. This can be explained by the fact that the model was determined by the remove/restore approach with respect to the EGM2008 geoid model [32], reducing the errors contained in the geoid signal, which is the largest contributor to the measured SSH [2]. NMBU18 also uses the remove/restore approach, but the results are much worse. Again, DTU21 performs better than NMBU18, especially in the inner part of the fjord. Compared to the previous three models, IC-2 MSS shows the best agreement with the tide gauges in terms of standard deviations (23.2 cm), and it generally fits them better than the Norwegian MSS model, NMBU18, because Sognefjorden is outside the domain of NMBU18.

Exceptions include tide gauges located at the mouth of the Sognefjorden (BULA and EKRE), where NMBU18 has more data since it covers a larger time period (2010–2018). The quality of IC-2 observations in this area might also be affected by the abundant precipitation in western Norway, which can also subsequently cause a lack of laser altimetry observations due to frequent cloud coverage. Even more evident height differences (>50 cm) can be observed at FJAE, AARD and SKJO. These three tide gauges sit at the innermost part of three different branches of the Sognefjorden: Fjærlandsfjorden (FJAE), Årdalsfjorden (AARD) and Lustrafjorden (SKJO).

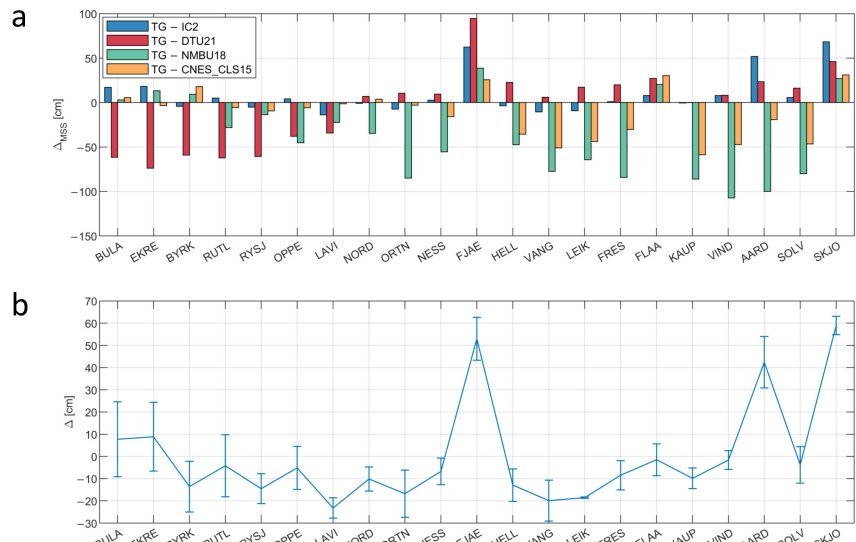

**Figure 6.** The height difference between temporary tide gauges and the IC-2 MSS model inside Sognefjorden (**a**). Signal standard deviations of the closest grid cell with available ICESat-2 observations (**b**).

**Table 4.** Statistics of differences between the MSL observed by temporary tide gauges inside Sognefjorden and MSS from altimetry.

| Δ | MIN [cm] | MAX [cm] | MEAN [cm] | STD [cm] |
|---|---|---|---|---|
| TG-IC2 | −13.8 | 68.3 | 9.4 | 23.2 |
| TG-DTU21 | −73.8 | 94.5 | −3.9 | 42.9 |
| TG-NMBU18 | −107.3 | 38.8 | −39.0 | 45.2 |
| TG-CNES_CLS15 | −58.7 | 31.1 | −12.5 | 27.4 |

Looking at the data distribution in these three tributary fjords (Figure 7), it is clear that the closest available observations are ∼15 km from each tide gauge. Thus, large differences can be explained by the uncertainties associated with the extrapolation of the MSS towards the inner end of the fjords. Nevertheless, this explanation can be elaborated with the values of the signal standard deviations of the nearest grid cell where we actually have data (Figure 6).

It shows that the accuracy of the existing data is not in question and that the standard deviations of the available observations within the Sognefjord are around 8 cm. This indeed confirms that ICESat-2 is able to retrieve more observations inside fjords than radar altimetry and that their accuracy is at a high level.

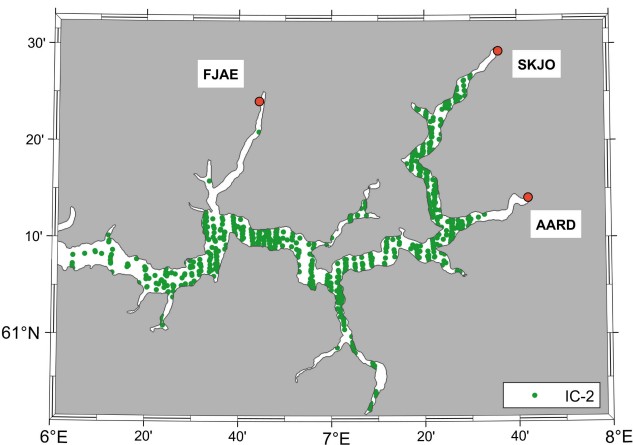

**Figure 7.** Branches of Sognefjorden with available ICESat-2 observations.

## 5. Conclusions

In this study, we have investigated the potential of ICESat-2 to improve existing MSS for Norway, especially along the coast and inside fjords. The coast of Norway was chosen because it has a very complex coastline with numerous deep fjords which are problematic to radar altimetry. ICESat-2 is NASA's laser altimetry mission equipped with the novel photon-counting Advanced Topographic Laser Altimeter System, which allows taking measurements every 0.7 m in a along-track direction, compared to SAR altimetry missions with an along-track spatial resolution of 300 m. Taking into account the characteristics of the Norwegian coast, previous studies by [18,20,39] have shown that both the quality and number of altimetry measurements can be severely degraded, especially inside fjords. Therefore, we determined the MSS model only from IC-2 observations, comparing it with the Norwegian network of permanent tide gauges along the entire coast and temporary tide gauges in Sognefjorden.

The mean sea surface model based only on ICESAT-2 observations shows lower standard deviations between tide gauges and IC-2 MSS (11.1 cm) compared to DTU21 (14.6 cm), NMBU18 (16.0 cm) and CNES_CLS15 (21.6 cm). The greatest improvement over the existing MSS model for Norway, NMBU18, is seen at the land-based tide gauges (NARV, TRON, OSCA and OSLO), where the quality of the radar altimetry observations is poor or there are no valid observations. Furthermore, when looking at the MSS performance inside the Sognefjord, IC-2 undoubtedly shows the best performance among all other models, with standard deviations of differences of 11.1 cm. The only exceptions are three tide gauges located at the innermost ends of the Sognefjord branches (FJAE, AARD and SKJO),where there is no ICESat-2 data nearby and uncertainties in extrapolation lead to larger differences between the between the IC-2 MSS model and these tide gauges.

Comparing the accuracy of the ICESat-2 observations along the whole coast (Table 3) and inside Sognefjorden (Table 4), it is clear that even within fjords the accuracy is not degraded. On the other hand, the accuracy of the closest MSS grid cell to each tide gauge can be significantly degraded at some stations due to different factors, e.g., the number of observations, and/or uncertainties in the interpolation method. This limitation opens possibilities for future studies which should address the optimum interpolation technique, especially when taking into account the characteristics of the Norwegian coast, as well as combination of ICESat-2 observations with radar altimetry observations to update the new mean sea surface model for Norway. The results show a large improvement in the coastal zone compared to other MSS models based only on radar altimetry observations. This points to the adequacy of the ICESat-2 utilization for improving the existing coastal mean sea surface.As a longer time series of ICESat-2 data becomes available, we can expect the influence of ocean variability to decrease and the accuracy of MSS models incorporating ICESat-2 observations to improve in the future. Along the coast of Norway we were fortunate to have a high-resolution ocean tide model covering all the fjords. In many other regions of the world, we do not have regional ocean tide models but only global ocean tide models, such as FES2014 or GOT4.8, which do not cover inside fjords. Here the use of ICESat-2 altimetry for MSS determination will be significantly much more problematic as we have to deal with both oceanographic, meteorological and tidal variability in sea level.

**Author Contributions:** Conceptualization, M.T. and O.B.A.; methodology, M.T. and O.B.A.; software, M.T.; validation, M.T.; formal analysis, M.T. and O.B.A.; investigation, M.T. and O.B.A.; resources, O.B.A.; data curation, M.T.; writing—original draft preparation, M.T.; writing—review and editing, M.T. and O.B.A.; visualization, M.T.; supervision, O.B.A.; project administration, M.T. and O.B.A. All authors have read and agreed to the published version of the manuscript.

**Funding:** This research was funded by the Norwegian University of Life Sciences under project number 651040.

**Data Availability Statement:** NASA's ICESat-2 ATL12 data can be downloaded from the NSIDC website: https://nsidc.org/data/atl12 (accessed on 9 May 2023). The tide gauge observations are available and distributed by the Norwegian Mapping Authority, Hydrographic Service (https:

//www.kartverket.no/en/api-and-data/tidal-and-water-level-data; accessed on 19 January 2023). DTU21MSS model can be downloaded from https://ftp.space.dtu.dk/pub/. CNES_CLS15 MSS model can be downloaded from https://www.aviso.altimetry.fr/en/ (accessed on 25 June 2023).

**Acknowledgments:** We would like to thank K. Breili at NMA for providing TG data and NMBU18 MSS model. We would also like to thank both K. Breili and C. Gerlach at the Bavarian Academy of Sciences and Humanities for reading the manuscript and providing helpful comments. NSDIC is acknowledged for providing ICESat-2 data. Figure 1 is reprinted from [9], Copyright (2023), with permission from Elsevier. This work is part of the Norwegian University of Life Science's SEGREF project, supported by the Norwegian University of Life Sciences under project number 651040. Finally, we would like to thank Reviewers for all valuable comments and suggestions, which helped us to improve the quality of the manuscript.

**Conflicts of Interest:** The authors declare no conflicts of interest.

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
