# Peer review of "ICESat-2 for Coastal MSS Determination—Evaluation in the Norwegian Coastal Zone"

_remotesensing, doi:10.3390/rs15163974_

Round 1
Reviewer 1 Report
All together well-preserved research, within very interesting topic. The biggest problem I have with this paper is that is not clear who this research would be useful for. Minor comments are in points 1-7 and stronger comments that need more work on them are in points 8 and 9. A goal and expected users should be explained in more clear manner.
1. Line 6, it should be put in neutral speech, instead “we wanted” it should be written the authors wanted, or it was expected or something neutral. It should be checked throughout the paper.
2. Please add some more keywords
3. Title is not good and clear
4. What is the source of Fig 1 and why is it set in Introduction?
5. Line 98 Fig ??
6. Line 110 and 112 URL instead of link
7. Line 138 25?
8. Line 231 The results seem promising – it is not clear what the authors mean by that.
9. The questions that should be raised by the paper are:
- who is it useful for, (for instance like stated in point 8 of this review);
- what are the weaknesses of the research;
- what could be future research expectations;
- what are the recommendations and implications of the research;
- how could/should futures studies improve the model etc.
Reviewer 2 Report
This paper present the evaluation of a new high-spatial resolution altimetry SSH product ICE-2 in representation the coastal and inland water level along the Norwegian coast and one fjord region. The new dataset was evaluated versus data from many tide stations in the region as well as several different SSH models based on previous altimeter measurements using lower-resolution sensors. It as concluded that the ICE-2 dataset, significantly improved SSH representation for the study region even inside the fjord area.
The paper is well-written with a reasonable outline. However, the paper lacks presentations of details about the method, data, and study area. I would like to recommend this paper as a candidate to be published in Remote Sensing after going through some major revisions. More details are presented below:
-More details on the remote sensing part: since the paper is going to appear in the ‘Remote Sensing’ journal, more details on the remote sensing part of the study including the satellite, measurement technique, data analysis of different levels, and the data tracks are needed. The followings are some examples;
· Line 86-88: samples of the 2-Gaussian data distributions and more details about the four moments should be presented
· A Schematic of data measurement by ICE-2 showing the emitted and returned back laser signals.
· Sample tracks of ICE-2 measurements at different times around the Scandinavian peninsula and adjacent waters in the North Sea
-More details on the hydrodynamics of sea level and met-ocean conditions in the study area are needed
Several factors including tide, storm surge, and wave setup can affect the water level in the coastal regions. In most cases, the combination of all these reasons contributes to the total water level. The reason that this information are important is that in this paper looks like the authors used the average SSH through time, while depending on the meteorological conditions, the time averages can change.
The author needs to add more details to the paper about each of them and how they contribute to the total water level in the study area. What is (in general) the minimum and maximum of tidal level with respect to the mean sea level? In Tables 1 and 2 column(s) should be added to show these values at each station.
How about the storm surges? The winter storms in the North Sea are severe and can cause very large storm surge along the Norwegian coasts. The authors need to elaborate more on its contribution to the total water level.
Depending on the water depth at the location of each tidal gauge, the effect of wave setup can be important in the total water level. The author needs to present the total water depot at each tidal gauge and discuss the effect of wave setup.
- As mentioned above, this is the reviewer’s understanding that the authors used the time-averaged SSH to compare ICE-2 data with other products (Figures 4 and 5). If this is the case, the author needs to address the following questions:
1. While the data span for the four data products other than ICE-2 is general from 1993-2012 or other different periods, the tide gauge data ate mostly from 2018-2022 (the data period used for TCE-2 was not in the paper). If the authors used time-averaged SSH for comparison, this time inconsistency can be problematic.
2. Since the return period of ICE-2 is 91 days, each year only four sets of data for the study area will be available. Not sure how many years of data are used for ICE-2, but it does not seem to be enough to make a time average of SSH. What if some of the data are corresponding to extreme meteorological conditions that are not representative of the average conditions in the region?
- The paper needs a more robust and extended discussion that addresses different types of inconsistencies between data and tide gauges and considers the met-ocean conditions of the study region.
Reviewer 3 Report
Review Report for the manuscript:
Coastal mean sea surface improvement with IceSat-2 in Norway
General:
In my opinion this manuscript looks good, it try to assess new satellite altimeter data for seawater surface measurement.
There are some few nots to improve the manuscript:
Please, use “ICESat-2” in all text and the title
The ICESat-2 has repeat cycle of 91 days, it is not clear how many cycles did you use and the comparison and uncertainty between ICESat-2 results and Tide gauge is not clear.
The equations of the vertical reference consistency between different data doesn’t mention (Ellipsoidal and Geiod data)
Figure 1. Should be for location of study area, depth legend missed.
Page 2: Line 54 to 67 were copied from the same reference [15].
What the difference between ATL12 and ATL03 photons, which used to assess heights of Trees, seawater surface. Please, mention the difference and importance of ATL12.
Page 3 line 91. Please capitalize dynamic ocean topography (DOT)
Page 3 Line 98. Fig missed
In the Table 1. Please, remove the columns start and end dates (they are the same) you can mention the data time in the title.
Page 4. Line 105. The calculation method should be inserted.
Page 5. Line 105. The conversion approach should be summarized in the context. Also in Line 113. The correction technique is missed.
Page 5.line 124. The period (1993-2012) is 19-years not 20-years.
Page 6. Line 142 please insert the datum expression transformation equations.
Page 6 Line 144. What is the source of The value 3.5 mm/yr. ?
Please, aware that there are spatial variations along grid cells.
Figure 3. The mean sea surface for Norway made from ICESat-2 When exactly.
As you know, it may be different between months/seasons.

Round 2
Reviewer 2 Report
I think the authors addressed all my concerns